# Unraveling the Paradox of Vitamin D Status in Primary Hyperparathyroidism: An Incidental Finding or an Unexpected Consequence?

**DOI:** 10.3390/ijms26094434

**Published:** 2025-05-07

**Authors:** Oriana-Eliana Pelineagră, Ioana Golu, Melania Balaș, Daniela Amzăr, Iulia Plotuna, Oana Popa, Mihaela Vlad

**Affiliations:** 12nd Department of Internal Medicine–Discipline of Endocrinology, “Victor Babes” University of Medicine and Pharmacy, P-Ta Eftimie Murgu 2, 300041 Timisoara, Romania; oriana.pelineagra@umft.ro (O.-E.P.); balas.melania@umft.ro (M.B.); amzar.daniela@umft.ro (D.A.); iulia.plotuna@umft.ro (I.P.); oana.taban@umft.ro (O.P.); vlad.mihaela@umft.ro (M.V.); 2Department of Endocrinology, County Emergency Hospital Timisoara, Blvd. Liviu Rebreanu 156, 300723 Timisoara, Romania; 3Center for Molecular Research in Nephrology and Vascular Disease, “Victor Babes” University of Medicine and Pharmacy, P-Ta Eftimie Murgu 2, 300041 Timisoara, Romania

**Keywords:** vitamin D deficiency, normocalcemic primary hyperparathyroidism, parathyroidectomy, nephrolithiasis, osteoporosis, parathyroid adenoma

## Abstract

Suboptimal vitamin D status is commonly observed in primary hyperparathyroidism but is rarely considered in management decisions. The present study aimed to bring additional insights on vitamin D status in primary hyperparathyroidism patients, particularly those presenting with the normocalcemic phenotype. A retrospective study was conducted on 53 confirmed primary hyperparathyroidism patients, stratified into hypercalcemic and normocalcemic groups, hospitalized at the “Pius Brînzeu” Emergency Clinical Country Hospital in Timișoara, Romania. Patients presenting with the normocalcemic phenotype had similar target-organ involvement compared to their counterparts. In this subgroup, 25 hydroxyvitamin D showed an inverse correlation with serum calcium (*p* = 0.048), and regression analysis identified iPTH and 25OH vitamin D as significant predictors of calcium levels (*p* < 0.0001; $R^{2}$ = 0.571). Adenoma volume showed a significant negative correlation with 25OH vitamin D levels (*p* = 0.021; r = −0.61) but was later found as insignificant after confounder analysis. Postoperative measurements of 25OH vitamin D levels confirmed increasing levels after parathyroidectomy. Our findings highlight a complex relationship between PTH and vitamin D in primary hyperparathyroidism, especially in the often-underdiagnosed normocalcemic phenotype. The inverse correlation between vitamin D and calcium suggests altered homeostasis, rather than true deficiency.

## 1. Introduction

Vitamin D has been long recognized as a pivotal regulator in bone health and calcium metabolism. Beyond its well-established effects on bone mineralization, vitamin D and its metabolites have emerged as key modulators for various endocrine, immune, and proliferative processes, highlighting their broader clinical significance. Despite 1,25 dihydroxyvitamin D (1,25${\left(OH\right)}_{2}$ vitamin D) exerting direct physiological effects at the tissue level, 25 hydroxyvitamin D (25OH vitamin D) remains the most commonly used biomarker for vitamin D status. Its synthesis commences with skin-mediated photoconversion of 7-dehydrocholesterol to cholecalciferol, followed by hepatic hydroxylation to 25OH vitamin D [1,2,3]. Although other extrahepatic tissues express 25-hydroxylase activity, their contribution to total plasma concentrations appears minor [4]. For vitamin D to exert its physiological role, a second hydroxylation process has to take place in the kidney [2]. 25OH vitamin D undergoes glomerular filtration and is reabsorbed back into the cells of the proximal convoluted tubule where 1α-hydroxylase catalyzes its conversion to 1,25${\left(OH\right)}_{2}$ vitamin D, the form with the highest affinity for the vitamin D receptor. The activity of 1α-hydroxylase is tightly regulated by parathyroid hormone (PTH), calcium and phosphate concentrations, fibroblast growth factor 23 (FGF-23), and, not lastly, serum 1,25${\left(OH\right)}_{2}$ vitamin D itself [5]. In addition, several extrarenal tissues, including the parathyroid glands, express 1α-hydroxylase activity although the regulatory mechanisms are believed to be tissue-specific [4,6]. 1,25${\left(OH\right)}_{2}$ vitamin D, and to a yet unknown extent other vitamin D metabolites, bind to the vitamin D receptor, inducing various gene transcriptions as well as non-genomic effects. Calcium reabsorption is enhanced in the proximal convoluted tubule as well as in the intestine by the upregulation of specific proteins [5]. Skeletal effects include enhanced bone matrix protein synthesis, increased mineralization through an elevated calcium–phosphate product as well as osteoblast differentiation and maturation [7].

PTH is secreted by the parathyroid glands in response to fluctuations of the serum calcium concentrations, mediated by the calcium-sensing receptor (CaSR) [8]. Similarly to vitamin D, PTH serves as a phosphocalciotropic hormone that enhances renal calcium reabsorption, promotes skeletal calcium mobilization, and indirectly increases intestinal calcium absorption via the stimulation of 1,25${\left(OH\right)}_{2}$ vitamin D synthesis [5,8].

Primary hyperparathyroidism is defined by autonomous PTH secretion by one or more abnormal parathyroid glands, which leads to increased serum calcium. This condition is associated with renal complications in the form of nephrolithiasis, nephrocalcinosis or polyuria, and enhanced bone resorption contributing to lowering bone mineral density and osteoporosis [9,10]. Traditionally, primary hyperparathyroidism was described as involving overt hypercalcemia and target-organ complications; however, recently recognized phenotypes include asymptomatic forms without renal or skeletal involvement or a normocalcemic phenotype characterized by elevated PTH and normal serum calcium in the absence of secondary causes [11,12,13].

Recent studies hypothesized different interrelations between increased PTH values and suboptimal vitamin D levels, proving that certain conditions can have overlapping features, creating confusion and delayed intervention. In particular, for patients with primary hyperparathyroidism, further research is required to establish whether low vitamin D levels in this population reflect true deficiency or are a result of altered hepatic protein synthesis and enhanced conversion, modulated by PTH [13,14]. The present study aims to offer additional insights regarding vitamin D levels in primary hyperparathyroidism, with a particular focus on normocalcemic primary hyperparathyroidism, a phenotype that is frequently misdiagnosed or overlooked. While suboptimal vitamin D levels are incidentally observed in the hypercalcemic variant of primary hyperparathyroidism (HPHPT) without impacting further management, vitamin D deficiency serves as an exclusion criteria for the diagnosis of normocalcemic primary hyperparathyroidism (NPHPT).

## 2. Results

A total of 53 patients (45 women and 8 men) diagnosed with primary hyperparathyroidism met the inclusion criteria and were included in this study. The cohort was stratified into two subgroups based on total serum calcium levels: HPHPT patients with total serum calcium > 10.3 mg/dL and NPHPT with total serum calcium < 10.3 mg/dL. Descriptive characteristics including mean tendency indicators and dispersion indicators were calculated for age, BMI, iPTH, 25OH vitamin D, total serum calcium, ionized serum calcium, serum phosphate, urinary calcium, urinary phosphate, serum creatinine, and T scores from DXA at the lumbar spine and hip (Table 1).

Of the 53 patients included in our study group, 26.4% were confirmed as NPHPT cases. When comparing the two subgroups, no significant differences were observed in terms of demographic characteristics such as age, gender, or BMI. As expected, laboratory measurements revealed significantly higher total serum calcium concentrations as well as higher ionized serum calcium concentrations in the HPHPT subgroup compared to the NPHPT patients. In terms of disease severity, objectified by iPTH and alkaline phosphatase values, our findings suggest a more active form of disease present in hypercalcemic patients paired with more elevated values of both laboratory parameters (*p* = 0.023 and *p* = 0.044 for iPTH, respectively, alkaline phosphatase). Other parameters including serum phosphate, urinary calcium, or urinary phosphate showed no significant differences between subgroups, although serum phosphate was noticeably lower in HPHPT individuals (*p* = 0.077). Notably, the serum concentrations of 25OH vitamin D were found to be significantly lower in the HPHPT subgroup compared to the normocalcemic cohort (15.7 vs. 20.5 ng/mL *p* = 0.010) (Figure 1).

Target-organ complications such as osteoporosis or nephrolithiasis were assessed by abdominal ultrasound and DXA. Despite the lower iPTH levels observed in our normocalcemic cohort, no significant differences were detected between the two subgroups regarding these complications. For bone mineral density, results were obtained and compared as T scores for lumbar spine (L1-L4) (*p* = 0.531) and total T score at the hip joint (*p* = 0.439). Similarly, the presence of nephrolithiasis was comparable between groups, supporting the abovementioned result of similar urinary calcium excretion between subgroups (*p* = 0.608). Non-classical symptoms related to hyperparathyroidism were recorded as total symptom count rather than categorized individually due to wide variability in patient-reported complaints. Reported symptoms included headache, cognitive/memory impairment, muscle pain/weakness, joint pain, polydipsia, polyuria, fatigue, and weigh loss. Non-classical manifestations were found to be more prevalent among HPHPT patients, with most reporting between one and three symptoms. In contrast, NPHPT patients presented with isolated symptoms or were entirely asymptomatic (*p* = 0.002).

To further investigate the relationship between 25OH vitamin D, normocalcemic status, and disease burden, a combined group analysis using Spearman’s rank correlation was conducted. A strong correlation was observed between iPTH values and total serum calcium levels (*p* = 0.0001) while 25OH vitamin D levels appear to be negatively correlated with calcium levels showing a weak but significant correlation (*p* = 0.048) (Figure 2 and Figure 3). Multiple regression analysis identified both 25OH vitamin D and iPTH levels as significant predictors for total serum calcium levels (*p* < 0.0001; $R^{2}$ = 0.571).

We further analyzed whether subjective disease burden through non-classical symptoms could be explained by individual calcium concentration, iPTH, or 25OH vitamin D levels. Total serum calcium demonstrated a moderate but significant correlation with the number of symptoms (*p* = 0.0015; r = 0.45). However, no significant correlation was found between disease burden and either iPTH (*p* = 0.148) or vitamin D status (*p* = 0.867). In the linear regression analysis, iPTH levels remained a statistically significant predictor of the number of non-classic symptoms although the determination coefficient suggested only a minor effect (*p* = 0.006; $R^{2}$ = 0.135). When all three factors were included in a multivariate model for their influence on symptomatology, total calcium levels remained the only parameter with a notable involvement (*p* = 0.007; $R^{2}$ = 0.215).

In the NPHPT group, statistical analysis was performed to determine factors influencing adenoma volume. Notably, the levels of 25OH vitamin D emerged as a predictor for adenoma volume, confirmed by Spearman’s rank correlation (*p* = 0.021; r= −0.61) and linear regression analysis (*p* = 0.037; $R^{2}$ = 0.366) (Figure 4). Another potential disease marker, alkaline phosphatase, demonstrated significant correlation with adenoma volume (*p* = 0.029; r = 0.62) and was identified as a significant predictor for adenoma volume in this subgroup (*p* = 0.041; $R^{2}$ = 0.384) (Figure 5). When alkaline phosphatase was included as a covariate in partial correlation with vitamin D for adenoma volume, 25OH vitamin D levels became statistically insignificant (*p* = 0.128), suggesting that the previously noted association might be confounded by other variables.

Finally, paired data from six patients were selected for further analysis if all laboratory measurements before and at least 6 months apart from surgery were available. Paired analysis of variables documented significantly lower iPTH (*p* = 0.031), total serum calcium (*p* = 0.008), and higher 25OH vitamin D levels (*p* = 0.003) after parathyroidectomy. As we suspected, decreasing iPTH values were influencing 25OH vitamin D levels, so we conducted correlation (*p* = 0.265; r = 0.543) and regression analysis (*p* = 0.098; $R^{2}$ = 0.535), which turned out to be not significant, probably in the context of a small sample.

## 3. Materials and Methods

An observational, retrospective study was conducted in our clinic by reviewing the medical records of the previously admitted primary hyperparathyroidism patients. All cases diagnosed with PHPT between January 2020 and January 2025 have been included for evaluation. Following the application of inclusion and exclusion criteria, a total of 53 patients were selected for further analysis, including 14 NPHPT patients. Patients with a confirmed diagnosis of multiple endocrine neoplasia syndrome or patients that had incomplete data regarding 25OH vitamin D status, serum calcium, iPTH (intact parathyroid hormone), or serum creatinine have been excluded. All causes for secondary hyperparathyroidism, such as medication use (diuretics, anticonvulsants, denosumab), renal impairment (eGFR < 60 mL/1.72 m^2^/min), vitamin D deficiency (25OH vitamin D < 20 ng/mL), or impaired calcium reabsorption have been excluded prior to a diagnosis of NPHPT [12]. On cervical ultrasound solid, hypoechoic lesions located posteriorly to the thyroid lobe, presenting with vascular arch, were suspected as parathyroid lesions and were further evaluated through 99Tc-sestamibi dual-phase scintigraphy. We chose to include patients with vitamin D levels below 20 ng/mL in the NPHPT group if both the abovementioned examinations were positive for parathyroid adenoma.

Collected data included demographic and anthropometric parameters such as age, gender, and body mass index (BMI), alongside laboratory assessments. Biochemical analyses comprised iPTH, total and ionized serum calcium, non-organic serum phosphate, alkaline phosphatase, serum creatinine, urinary calcium, urinary phosphate, and 25-hydroxyvitamin D (25OH vitamin D). Serum iPTH was measured using chemiluminescence on the Atellica Solution CH1 system (normal range: 13.6–85.8 pg/mL). Total serum calcium (normal range: 8.3–10.3 mg/dL), ionized calcium (normal range: 4.2–5.2 mg/dL), serum phosphate (normal range: 2.4–5.1 mg/dL), alkaline phosphatase (normal range: 38–126 U/L), serum creatinine (normal range: 0.52–1.04 mg/dL), urinary calcium (normal range: 42–353 mg/24 h), and urinary phosphate (normal range: 0.4–1.3 g/24 h) were determined via spectrophotometry on the Atellica Solution CH1 system. Serum 25OH vitamin D levels were measured using chemiluminescence on the Advia Centaur XPT analyzer.

Findings from localization imagining by US, 99Tc-sestamibi dual-phase scintigraphy, magnetic resonance imaging (MRI), or computed tomography (CT) scans were reviewed and lesion dimensions were recorded. Cervical ultrasound was performed using an Esaote MyLab Six ultrasound machine equipped with a 3–13 MHz linear transducer (SL1543). Target-organ involvement was assessed by abdominal ultrasound for nephrolithiasis and by Dual-energy X-ray Absorptiometry (DXA) for osteoporosis. Bone mass density was evaluated by the T score of the lumbar spine, total hip, and femoral neck. Localization imaging as well as the abovementioned screening assessments were labeled either as positive or negative. Lesion volume was calculated using a standardized formula (volume [${cm}^{3}]$ = π/6 × a × b × c) based on the three dimensions obtained from the histopathological report if available or, alternatively, dimensions of the lesion recorded during conventional cervical ultrasound.

Statistical analysis was performed using MedCalc^®^ Statistical Software version 23.1.1. Subgroup analysis was followed by comparison between the two groups using the unpaired Student *t*-test or Mann–Whitney U test for numerical variables, as appropriate. For nominal variables such as gender or presence of nephrolithiasis, Chi-Square test or Fisher exact test were used to assess significant differences between groups. Correlation between variables was investigated using Spearman’s rank correlation. Differences with *p* < 0.05 were considered statistically significant.

## 4. Discussion

In our cohort, we found that NPHPT patients present with similar complications compared to their HPHPT counterparts. Notably, we observed no difference in terms of T scores on DXA or nephrolithiasis prevalence despite hypercalcemic patients having elevated serum iPTH levels and alkaline phosphatase, suggestive for an enhanced bone turnover. Within our study, 15.4% of NPHPT patients presented nephrolithiasis on abdominal ultrasound, while 21.5% of the HPHPT cohort were similarly affected. The comparable prevalence of nephrolithiasis may be attributed to urinary calcium levels as this parameter displayed no significant difference between subgroups. Previous studies highlighted this aspect as well with a similar reported percentage of stone formers among NPHPT patients [15,16]. The prevalence of nephrolithiasis in NPHPT is variable, some authors reporting numbers as high as 29% [17]. The true incidence of nephrolithiasis in NPHPT patients remains rather unclear due to an atypical presentation of these patients, who are rather diagnosed accidentally when they are investigated for recurrent nephrolithiasis or osteoporosis, as opposed to HPHPT patients that are diagnosed after the routine measurement of serum calcium and are otherwise asymptomatic. This diagnostic discrepancy leads to a difficult estimation in disease prevalence and its complications rate [11].

Primary hyperparathyroidism patients have been shown to have lower vitamin D levels compared to matched controls, raising the question whether this population is truly vitamin D-deficient [14]. Vitamin D status can be classified based on serum 25OH vitamin D levels as the following: sufficient (>30 ng/mL), insufficient (20–30 ng/mL), or deficient (<20 ng/mL) [12]. Declining vitamin D levels are often accompanied by a compensatory increase in PTH, primarily aimed at preserving calcium homeostasis [18]. Suboptimal vitamin D levels have been previously reported in patients with primary hyperparathyroidism while underlying pathophysiological mechanisms remain a subject of debate [13,19,20]. Our current findings align with previous reports, documenting a mean 25OH vitamin D level of 15.7 in the HPHPT cohort and a higher level of 20.57 ng/mL in the NPHPT subgroup. Despite normocalcemic patients displaying higher vitamin D levels, as per the most recent classification, they are still considered as having an insufficient level. Vitamin D levels in the NPHPT population are rarely reported as low levels represent an exclusion criteria for most patients presenting with normal calcium and elevated iPTH levels. We chose to include this subset of patients if a parathyroid lesion could be objectified on cervical ultrasound, computer tomography, or MRI. Various studies reported conflicting values for total 25OH vitamin D and total 1,25${\left(OH\right)}_{2}$ vitamin D in primary hyperparathyroidism. Some authors proposed that other metabolites such as free or bioavailable 25OH vitamin D or even free 1,25${\left(OH\right)}_{2}$ vitamin D as better surrogates for vitamin D status [21,22].

Recent publications investigated the possible mechanism underlying low vitamin D levels in hyperparathyroidism, further hypothesizing a cumulative effect of multiple factors [20]. A well-established pathophysiological mechanism relies on the renal hydroxylation of 25OH vitamin D to its more active metabolite 1,25${\left(OH\right)}_{2}$ vitamin D. PTH binds to its receptor in the proximal convoluted tubule stimulating the renal 1α-hydroxylase and concurrently inhibiting the 24-hydroxylase. This mechanism has the ability to increase active vitamin D, playing a beneficial role in maintaining homeostasis. In situations with unsuppressed PTH secretion like a parathyroid adenoma, this mechanism becomes detrimental, leading to enhanced calcium reabsorption and maintaining hypercalcemia [5]. Alternative mechanisms include declining concentrations of binding protein for 25 OH vitamin D and 1,25${\left(OH\right)}_{2}$ vitamin D. Vitamin D can be transported in the plasma in three different forms: as free 25OH vitamin D (accounting for <1% of total), bound to albumin (~10–15%), or bound to D-binding protein (DBP) (85–93%) [1,2,23]. DBP is known to be the predominant transport protein for 25OH vitamin D followed by albumin. Studies reported decreased levels of DBP and albumin in patients with primary hyperparathyroidism and an inverse linear correlation between iPTH and DBP or albumin levels [2,20]. Despite a reduction in the total 25OH vitamin D levels, patients tend to retain normal 1,25${\left(OH\right)}_{2}$ vitamin D levels and normal levels of free 25OH vitamin D in this context [1,22,24,25]. While theoretically suboptimal vitamin D levels should accompany low–normal calcium levels, in our cohort, we found an unusual inverse correlation between serum calcium and 25OH vitamin D levels in both the NPHPT and HPHPT patients. Our findings support the abovementioned reports of the enhanced conversion of 25OH vitamin D or decreased binding protein contributing to low total 25OH vitamin D levels rather than true deficiency. This distinctive feature in vitamin D metabolism is essential to consider, particularly in NPHPT patients where the levels of 25OH vitamin D are used as an exclusion criteria. Further review of the relationship between the vitamin D level and disease burden through non-classical symptoms found no correlation between vitamin D status or iPTH levels and patients’ complaints. Total serum calcium remained the main true predictor for the number of symptoms.

The authors reported lower vitamin D levels as negatively correlated with adenoma weight, signifying an added growth stimulus in case of suboptimal levels [26,27]. Despite the influence of vitamin D levels on parathyroid adenoma weight being previously reported, in our study, we documented a minor influence of 25OH vitamin D levels on adenoma volume in the NPHPT group, an observation that became statistically insignificant when the analysis was adjusted for confounding variables.

When the analysis was performed on paired data for patients that were referred to surgery, we identified significantly higher total 25OH vitamin D values after parathyroidectomy associated with normalized iPTH and serum calcium. Although we evaluated whether changes (calculated delta values) in iPTH could be used as a predictor for changes in vitamin D levels after surgery, regression analysis came out as not significant (*p* = 0.098; $R^{2}$ = 0.535), probably due to a small sample. Other groups investigated the effects of parathyroidectomy on 25OH vitamin D levels and found that patients display noticeable increases in 25OH vitamin D levels, its metabolites, and DBP after parathyroidectomy coupled with decreasing iPTH levels [27]. Our results are in line with the aforementioned results and suggest that vitamin D metabolism can be altered in hyperparathyroidism through mechanisms that need further research.

Study limitations include a relatively small sample size, primarily attributable to incomplete datasets, missing parameters, and ambiguous patient histories and follow-up information. In our cohort, only 14 patients were diagnosed with NPHPT. Notably, postoperative comparisons were confined to six cases, as several patients either underwent surgery at external centers or lacked postoperative 25OH vitamin D measurements. Furthermore, the absence of data on free 25OH vitamin D or DBP-bound vitamin D precludes a more nuanced understanding of the interplay between PTH levels and vitamin D metabolism. The statistical analysis could have been influenced by sample size, highlighting the need for larger cohorts in prospective studies to establish variables’ interplay in the postoperative setting. Additionally, external factors including sun exposure, dietary intake, and prior supplementation with vitamin D could not be accounted for due to the retrospective nature of the study.

Given the complex interplay between vitamin D metabolism and parathyroid function highlighted by our findings, future studies should focus on larger cohorts of NPHPT patients, incorporating comprehensive analysis not only of total 25OH vitamin D levels but also its free fraction and binding proteins such as DBP or albumin. Furthermore, evaluations of 1,25${\left(OH\right)}_{2}$ vitamin D or 1,25${\left(OH\right)}_{2}$/25 OH vitamin D ratios, in conjunction with these fractions, could provide deeper insights into the pathophysiology and disease progression in this distinct phenotype.

## 5. Conclusions

Our study provides further insights regarding the complex interplay between PTH and vitamin D in primary hyperparathyroidism, with a particular emphasis on the normocalcemic phenotype, which remains prone to underdiagnosis and misclassification. Despite maintaining normal serum calcium levels, patients diagnosed as NPHPT exhibit comparable prevalence of disease-related complications to their hypercalcemic counterparts. These findings reinforce previous studies that considered NPHPT as a separate phenotype rather than a milder form of hyperparathyroidism. Moreover, the significantly higher vitamin D levels reported in primary hyperparathyroidism patients after surgery suggest a complex interrelationship between parathyroid pathology and suboptimal vitamin D levels. The inverse correlation between vitamin D levels and serum calcium further supports the hypothesis that reduced vitamin D levels in PHPT might not reflect true deficiency but rather an altered homeostasis caused by parathyroid activity. Our current findings point toward suboptimal vitamin D status being an adaptive response to parathyroid-driven disruptions in calcium–phosphate metabolism, rather than an unrelated finding. These observations underscore the need for future research to delineate the underlying mechanisms governing vitamin D metabolism in this setting, which may ultimately refine diagnostic criteria and inform therapeutic strategies for both normocalcemic and hypercalcemic PHPT.

## Figures and Tables

**Figure 1 ijms-26-04434-f001:**
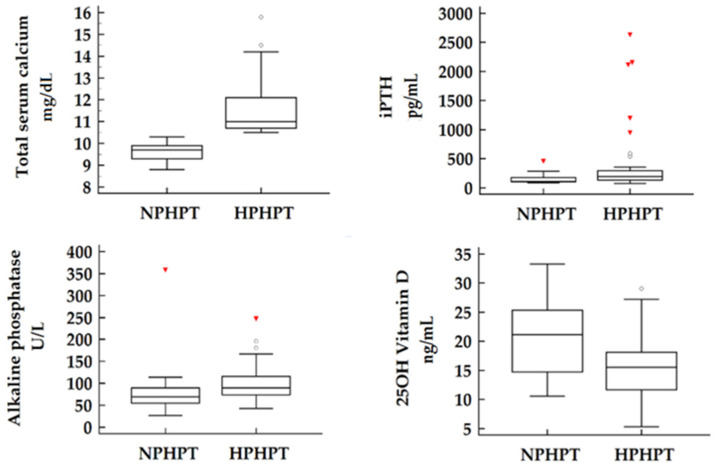
Laboratory assessments in HPHPT and NPHPT subgroups.

**Figure 2 ijms-26-04434-f002:**
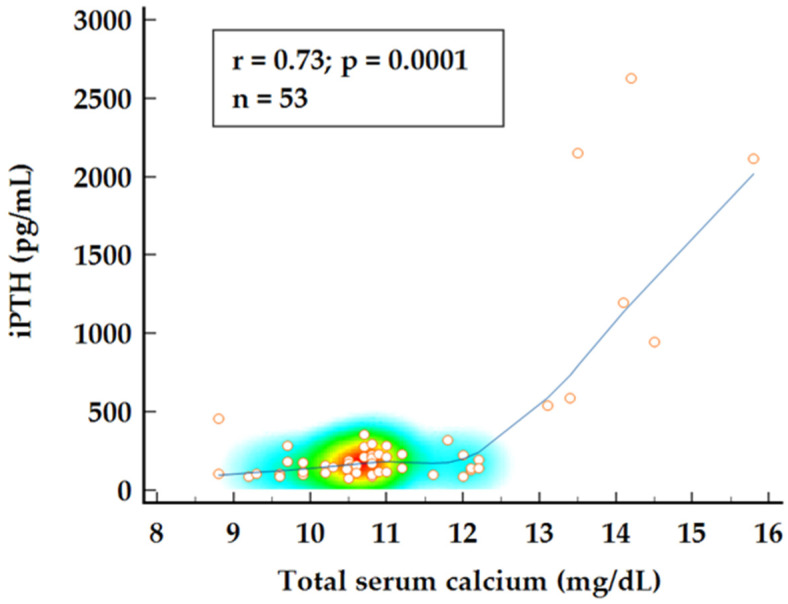
Spearman’s rank correlation for total serum calcium and iPTH levels.

**Figure 3 ijms-26-04434-f003:**
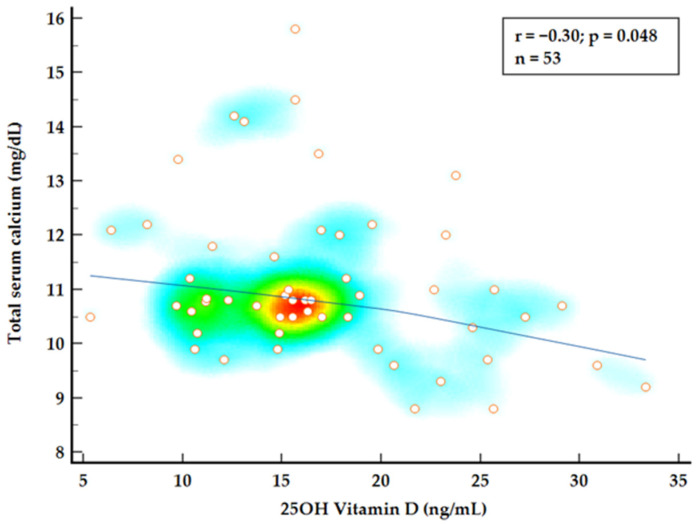
Spearman’s rank correlation for total serum calcium and 25OH vitamin D levels.

**Figure 4 ijms-26-04434-f004:**
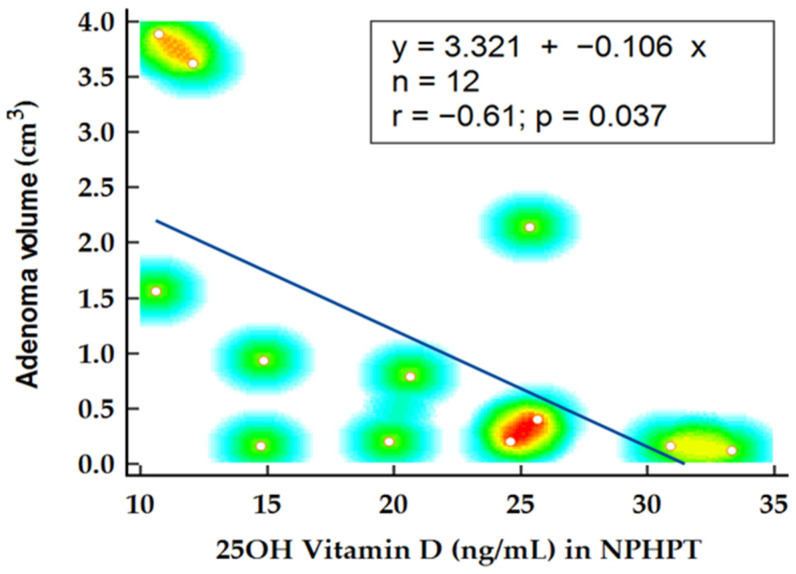
Linear regression analysis on adenoma volume and 25OH vitamin D levels.

**Figure 5 ijms-26-04434-f005:**
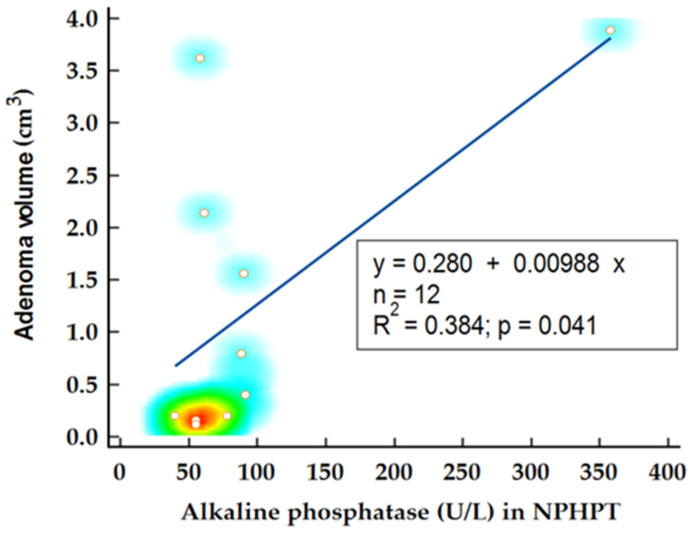
Linear regression analysis on adenoma volume and alkaline phosphatase levels.

**Table 1 ijms-26-04434-t001:** Subgroup characteristics and laboratory assessments.

Variable	HPHPT Patients (n = 39)	NPHPT Patients (n = 14)	*p*-Values
Age (years)	60 (53–68)	67 (60–71)	*p* = 0.060
Gender (F/M)	33/6	12/2	*p* = 1
BMI (kg/$m^{2}$)	29.46 ± 5.64	29.10 ± 5.49	*p* = 0.842
Total serum calcium (mg/dL)	11 (10.7–12.1)	9.7 (9.3–9.9)	** *p * ** **< 0.0001**
Ionized serum calcium (mg/dL)	5 (4.69–5.36)	4.5 (4.26–4.63)	** *p * ** **= 0.0001**
Serum phosphate (mg/dL)	2.74 ± 0.51	3.07 ± 0.72	*p* = 0.077
Serum creatinine (mg/dL)	0.80 (0.7–1.1)	0.76 (0.6–0.9)	*p* = 0.313
iPTH (pg/mL)	195 (135.7–296)	111.5 (105–178)	** *p * ** **= 0.023**
Alkaline phosphatase (U/L)	90 (74–116)	61 (55–90.25)	** *p * ** **= 0.044**
Urinary 24 h calcium (mg/24 h)	363.1 (234.4–447)	278 (117.3–372.8)	*p* = 0.173
Urinary 24 h phosphate (g/24 h)	0.7 (0.53–0.87)	0.77 (0.42–0.9)	*p* = 1
25OH vitamin D (ng/mL)	15.7 ± 5.3	20.57 ± 7.2	** *p * ** **= 0.0108**
Lumbar spine T score (standard deviations)	−2.33 ± 1.46	−2.68 ± 1.56	*p* = 0.531
Hip T score (standard deviations)	−1.53 ± 1.12	−1.22 ± 0.94	*p* = 0.439
Presence of nephrolithiasis	21.4%	15.4%	*p* = 0.608
No. of non-classic symptoms related to hyperparathyroidism	2 (1–3)	0 (0–1)	***p * = 0.0021**
Adenoma volume (${cm}^{3}$)	0.64 (0.19–6.7)	0.59 (0.18–1.85)	*p* = 0.686

n, number of patients; BMI, body mass index; iPTH, intact parathyroid hormone; 25OH vitamin D, 25-hydroxyvitamin D; data shown as mean ± standard deviation or median (IQR). *p* values considered statistically significant (<0.05) were written in bold.

## Data Availability

Due to the retrospective study design, patients included in this study did not give written consent for their data to be shared publicly, so due to the sensitive nature of the research, supporting data are not available.

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
