# Peer review of "Unraveling the Paradox of Vitamin D Status in Primary Hyperparathyroidism: An Incidental Finding or an Unexpected Consequence?"

_ijms, 2025, doi:10.3390/ijms26094434_

Round 1

Reviewer 1 Report

Comments and Suggestions for Authors

Pelineagră et al. conducted a retrospective study to provide additional insight into vitamin D status in patients with primary hyperparathyroidism, particularly those with the normocalcemic phenotype.  Their findings suggest a complex relationship between PTH and vitamin D in primary hyperparathyroidism, especially in the often underdiagnosed normocalcemic phenotype. I have a few comments about the study:

  • Spaces are missing in some places - e.g. lines 52-64; 263-270
  • Line 158: …(p=0.023 and p=0.044 for iPTH, respectively alkaline phosphatase)
  • I also suggest improving the readability of Figure 1
  • In Figure 4 r=0.61, but in the text it is r=-0.61 (line 209)
  • Why is the graph for alkaline phosphatase not shown?

Author Response

Pelineagră et al. conducted a retrospective study to provide additional insight into vitamin D status in patients with primary hyperparathyroidism, particularly those with the normocalcemic phenotype.  Their findings suggest a complex relationship between PTH and vitamin D in primary hyperparathyroidism, especially in the often underdiagnosed normocalcemic phenotype. I have a few comments about the study:

  1. Spaces are missing in some places - e.g. lines 52-64; 263-270

Response: We appreciate your careful reading of the manuscript. As noted, spaces between words and corresponding abbreviations or references were missing in several places despite careful rereading. These were corrected across the manuscript. Additionally, we modified  “1,25(OH)2 …” to “1,25…” in Line 67 to maintain consistency, as this notation form was omitted in previous versions.

  1. Line 158: …(p=0.023 and p=0.044 for iPTH, respectively alkaline phosphatase)

Response: The spacing issues in the sentence were corrected as suggested.

  1. I also suggest improving the readability of Figure 1

Response: Thank you for this valuable suggestions. All figures initially included in Figure 1 have been reworked with increased font size and remerged to improve readability. Additionally, intact parathyroid hormone has been abbreviated as iPTH in the figure to improve visual clarity.

  1. In Figure 4 r=0.61, but in the text it is r=-0.61 (line 209)

Response: Thank you for identifying this inconsistency. The correct correlation coefficient is r= -0.61 as stated in the text, the graphic figures legend was updated accordingly to display the correct value.

  1. Why is the graph for alkaline phosphatase not shown?

Response: We appreciate your observation about the graphic figure regarding alkaline phosphatase. The corresponding figure has now been included as Figure 5 and the text has been updated to indicate this addition. Line 215 – in the revised manuscript.

Manuscript revision was performed using Word’s “Track changes” feature to help reviewers in identifying modified lines. To avoid possible discrepancies between authors and reviewers line number in current manuscript version, revised phrases have been also written in red.

Reviewer 2 Report

Comments and Suggestions for Authors

The study by  Oriana-Eliana Pelineagră et.al. is interesting to read and well written. It provides an insight into the relations between the levels of vitamin D and hyperparathyroidism.

My comments:

  1. The title section includes a question: “(…) vitamin D status (…) primary hyperparathyroidism - an incidental finding or an unexpected consequence?” and I was hoping to find an answer to it in the conclusion section.
  2. Line 98 – 101 – did the authors use any particular guidelines for the threshold values described in this fragment? Especially concerning the 25OH vitamin D threshold of <20 100 ng/ml – I have found the answer to my question in the discussion section in line 246 – the authors should add reference 13 to line 101.
  3. Line 113 - serum phosphate (normal range: 2.4–5.1 mg/dL) – is this non-organic phosphate?
  4. The conclusion section: in my opinion the statement that “further research is needed…” should be removed from this section (especially because the discussion section already includes a similar sentence). It makes the impression that the results of the study are inconclusive

Author Response

The study by  Oriana-Eliana Pelineagră et.al. is interesting to read and well written. It provides an insight into the relations between the levels of vitamin D and hyperparathyroidism.

My comments:

  1. The title section includes a question: “(…) vitamin D status (…) primary hyperparathyroidism - an incidental finding or an unexpected consequence?” and I was hoping to find an answer to it in the conclusion section.

Response: We appreciate your thoughtful suggestion regarding the alignment between the title of the manuscript and the conclusion. While several points in the discussion section  highlight suboptimal vitamin D levels as an adaptive change in settings of primary hyperparathyroidism, we recognize the need to articulate it more clearly. We have revised the final portion of the discussion section to more explicitly support the interpretation that vitamin D status is a consequence – rather than an incidental finding – in parathyroid pathology. (Line 322-326 in the revised manuscript)

  1. Line 98 – 101 – did the authors use any particular guidelines for the threshold values described in this fragment? Especially concerning the 25OH vitamin D threshold of <20 100 ng/ml – I have found the answer to my question in the discussion section in line 246 – the authors should add reference 13 to line 101.

Response: Thank you for identifying this oversight. The corresponding reference has been added as suggested in Line 103 of the revised manuscript.

  1. Line 113 - serum phosphate (normal range: 2.4–5.1 mg/dL) – is this non-organic phosphate?

Response: Yes, all reported data for serum phosphate refer to non-organic phosphate. To ensure clarity for reader, we have revised the description of biochemical analyses in Line 110 of the revised manuscript. 

  1. The conclusion section: in my opinion the statement that “further research is needed…” should be removed from this section (especially because the discussion section already includes a similar sentence). It makes the impression that the results of the study are inconclusive.

Response: We appreciate your observation regarding this specific formulation. While our findings do underscore the complex interplay between vitamin D levels and parathyroid activity in both phenotypes and in a limited number of postoperative patients, we also aim to highlight the importance of comprehensive evaluation of vitamin D metabolism and  parathyroid function in future research to better describe the complex molecular mechanisms facilitating this changes. We fully recognize the importance of framing our conclusion in a way to avoid suggesting inconclusiveness. As a result, the conclusion section was revised to more clearly reflect the significance of our findings, while still highlighting aspects for future research without diminishing our study’s contribution. (Line 344-350 in the revised manuscript)

Manuscript revision was performed using Word’s “Track changes” feature to facilitate the reviewers identification of all modifications. To minimize potential discrepancies between authors and reviewers line number, revised phrases have been written highlighted in red in the current manuscript version.